# MetaWise: Combined Feature Selection and Weighting Method to Link the Serum Metabolome to Treatment Response and Survival in Glioblastoma

**DOI:** 10.3390/ijms252010965

**Published:** 2024-10-11

**Authors:** Erdal Tasci, Michael Popa, Ying Zhuge, Shreya Chappidi, Longze Zhang, Theresa Cooley Zgela, Mary Sproull, Megan Mackey, Heather R. Kates, Timothy J. Garrett, Kevin Camphausen, Andra V. Krauze

**Affiliations:** 1Radiation Oncology Branch, Center for Cancer Research, National Cancer Institute, National Institutes of Health (NIH), Bethesda, MD 20892, USA; erdal.tasci@nih.gov (E.T.); michael.popa@nih.gov (M.P.); zhugey@mail.nih.gov (Y.Z.); shreya.chappidi@nih.gov (S.C.); longze.zhang@nih.gov (L.Z.); theresa.cooleyzgela@nih.gov (T.C.Z.); sproullm@mail.nih.gov (M.S.); mmackey@mail.nih.gov (M.M.); camphauk@mail.nih.gov (K.C.); 2Department of Pathology, Immunology and Laboratory Medicine, University of Florida, Gainesville, FL 32610, USA; hkates@ufl.edu (H.R.K.); tgarrett@ufl.edu (T.J.G.)

**Keywords:** glioblastoma, metabolomic, compound, feature selection, machine learning, pattern recognition

## Abstract

Glioblastoma (GBM) is a highly malignant and devastating brain cancer characterized by its ability to rapidly and aggressively grow, infiltrating brain tissue, with nearly universal recurrence after the standard of care (SOC), which comprises maximal safe resection followed by chemoirradiation (CRT). The metabolic triggers leading to the reprogramming of tumor behavior and resistance are an area increasingly studied in relation to the tumor molecular features associated with outcome. There are currently no metabolomic biomarkers for GBM. Studying the metabolomic alterations in GBM patients undergoing CRT could uncover the biochemical pathways involved in tumor response and resistance, leading to the identification of novel biomarkers and the optimization of the treatment response. The feature selection process identifies key factors to improve the model’s accuracy and interpretability. This study utilizes a combined feature selection approach, incorporating both Least Absolute Shrinkage and Selection Operator (LASSO) and Minimum Redundancy–Maximum Relevance (mRMR), alongside a rank-based weighting method (i.e., MetaWise) to link metabolomic biomarkers to CRT and the 12-month and 20-month overall survival (OS) status in patients with GBM. Our method shows promising results, reducing feature dimensionality when employed on serum-based large-scale metabolomic datasets (University of Florida) for all our analyses. The proposed method successfully identified a set of eleven serum biomarkers shared among three datasets. The computational results show that the utilized method achieves 96.711%, 92.093%, and 86.910% accuracy rates with 48, 46, and 33 selected features for the CRT, 12-month, and 20-month OS-based metabolomic datasets, respectively. This discovery has implications for developing personalized treatment plans and improving patient outcomes.

## 1. Introduction

Glioblastoma, known as glioblastoma multiforme (GBM), is the most aggressive and common type of cancer originating in the brain [1,2]. It arises from astrocytes, star-shaped cells that support nerve cells. While there is no known prevention method, treatment typically involves maximal safe resection followed by concurrent chemotherapy (CRT), with temozolomide (TMZ) and radiation therapy (RT), followed by adjuvant chemotherapy [2,3,4,5]. The current standard of care (SOC) has remained largely unchanged for nearly 20 years, resulting in a median overall survival (OS) of 20 months in patients treated in clinical trials, and as low as 14 months inon real-world data. Despite treatment efforts, glioblastoma has a poor prognosis, with a median survival duration of 12 months after diagnosis, and fewer than 5–10% of patients surviving beyond five years [6]. GBM has nearly universal recurrence, driven by a number of cancer hallmark pathways including metabolic reprogramming that evade the damage induced by TMZ and RT. Different studies emphasize the importance of molecular diagnosis in personalizing glioma treatment, and combining molecular insights with these new techniques can enhance the effectiveness of glioma treatments for patients [7,8]. The most important genes influencing GBM tumor growth consist of EGFR, PTEN, TP53, IDH1, and MGMT mutations [7]. Although molecular classifications, such as the MGMT methylation status and IDH mutation, have improved our ability to perform prognosis, they have not resulted in personalized management. Currently, no biomarker is known to predict individuals’ response or resistance to the SOC.

Metabolomics holds great promise for analyzing GBM tumors and exploring potential treatment methods. Metabolomics offers a significant advantage by accurately reflecting the metabolic signature of a tumor’s phenotype and capturing the external factors that genomics or proteomics cannot precisely capture within analyzed cells [9]. A compound is a general term used in metabolomics to describe a substance composed of two or more different elements chemically bonded together [10]. Studying the metabolome before and after chemoirradiation and the overall survival status in GBM provides valuable insights into the tumor’s biochemical response, helps to identify biomarkers for treatment response and resistance, can guide the optimization of therapeutic strategies, and can improve disease management [11]. A comprehensive understanding of these mechanisms ultimately contributes to personalized medicine approaches and improved patient outcomes.

Large-scale metabolome analysis is increasingly used to investigate the signaling pathways associated with tumor resistance, with findings highlighting a significant interplay between metabolism and the resistance of cancer to treatment. The noninvasive analysis of the metabolome using serum samples from patients with GBM collected prior to and following the completion of CRT is novel, as is the acquisition of large-scale data in this field. Although this type of data is challenged by high dimensionality and evolving biological annotations, feature engineering can help uncover the metabolic strategies GBM uses to evade treatment, leading to recurrence and poor survival outcomes.

Feature selection using metabolomic data holds the potential to significantly enhance biomarker discovery and improve patient outcomes by employing robust dimensionality reduction techniques. The hidden patterns can be figured out and the prediction performance can be improved for specified tasks and datasets. Feature selection, as a critical preprocessing step, reduces data complexity by eliminating irrelevant data, accelerating the computation process, and facilitating data visualization [4,5,12]. This process can be categorized into filter, wrapper, and embedded methods based on the variable evaluation metrics [12]. In addition to feature selection, assigning appropriate weights to features through feature weighting is essential for optimizing the final feature subset identification in the cross-validation stage [4,5].

In this study, we focus only on the selection of metabolomic data-based biomarkers from large-scale metabolomic panels to apply our methodology to the specific data type and operations. We employed serum samples from GBM patients that were collected following surgical resection and prior to, as well as following, CRT. We aimed to test the use of combined feature selection and feature weighting in identifying the metabolomic compounds altered with treatment and associated with survival. To identify the most critical factors influencing CRT and the overall survival statuses within our local metabolomic dataset, we employed a feature selection and weighting approach (MetaWise) for high-dimensional serum metabolomic data to reduce it into a minimal set of informative and distinctive biomarkers. By enhancing the accuracy of model predictions, this research aims to optimize treatment strategies and improve the clinical outcomes o in patients with GBM.

The core contributions of this study, categorized as technical and clinical aspects, are presented below.

### 1.1. Technical Aspects

As far as we know, this is the first study to employ a hybrid feature (i.e., compound) selection and weighting methodology (i.e., MetaWise) for both pre- and post-CRT, 12-month OS, and 20-month OS status-based classification tasks on large-scale metabolomic data.To increase the scope and motivation of this study, we apply our approach to three different case studies using metabolomic data related to overall survival and biomarkers for the management of CRT.We adopted the ProFWise [4] methodology with an additional ensemble classification model (i.e., voting) and its parameters to extend our previous methodology to different types of omics data.To address the effects of imbalanced class distribution in our datasets, we employed stratified cross-validation during model training. This ensured that each training fold maintained the original class proportions.To determine the final feature names from the potential variations across cross-validation folds, we employed a rank-based feature weighting approach.We evaluated the effects of feature selection and weighting on six different machine learning models, using large-scale metabolomic datasets to determine the optimal prediction model and minimal feature set for accurate classification.We analyzed the feature alterations between pre- and post-CRT, the 12-month OS status, and the 20-month OS status.

### 1.2. Clinical Aspects

Serum samples representing noninvasive biospecimen collection in patients with histopathologically proven GBM were obtained prior to and following CRT, stored and defrosted for large-scale metabolomic data analysis, and linked to 12-month and 20-month OS. This was previously not possible, highlighting the use of novel large-scale metabolomic data linked to clinical outcomes.There are currently no known metabolomic biomarkers for GBM. The current approach, combined with interpretable dimensionality reduction (i.e., feature selection), allowed the identification of metabolic compounds with the high accuracy that discriminates pre-CRT from post-CRT and is associated with 12-month OS and 20-month OS. The serum metabolome can detect significant changes in signals between groups (pre- vs. post-treatment, survival time points), offering promising potential for future studies.

The remainder of this paper is structured as follows. Section 2 outlines the experimental setup, performance indicators, and computational outcomes in detail. Section 3 presents the results and discussion. Section 4 provides a detailed description of the dataset, the feature selection and weighting methodologies employed, and the supervised learning models used for classification. Finally, Section 5 summarizes the study’s findings and suggests potential directions for future research.

## 2. Results

The experimental setup and evaluation measures are described in this section, followed by a detailed presentation of the computational findings.

### 2.1. Experimental Process

To implement the proposed methods, we utilized Python’s scikit-learn [13] library for machine learning algorithms and the Minimum Redundancy–Maximum Relevance (mRMR) [14] package for filter-based feature selection operations. The Matlab (R2022b) environment was also used for visualization purposes.

All experiments were performed on a macOS Ventura MacBook Pro (2.3 GHz 8-core Intel Core i9, 16 GB of 2667 MHz DDR4 RAM). To achieve optimal results, this study employed six diverse predictive models, including the Support Vector Machine (SVM), K-Nearest Neighbors (KNN), Logistic Regression (LR), Adaptive Boosting (AdaBoost), Random Forest (RF), and the voting ensemble model, which were used in both feature selection and classification. To provide standardization for our previous similar studies [4,5] regarding different data types such as molecular, proteomic, and metabolomic data, and because there are many combinations to obtain from different feature subsets, datasets, classification models, and total or rank-ordered weights for cross-validation, we adopted the same and default parameter settings from Tasci et al. [4] for the feature selection and classification processes. The voting ensemble model used five prediction models, including LR, SVM, KNN, RF, and AdaBoost, by applying the soft voting rule. We also used an alpha parameter value of 0.01 for the Least Absolute Shrinkage and Selection Operator (LASSO) feature selection (FS) method used for the 20-month OS status-based dataset due to the features being highly correlated. To manage randomness and ensure consistent results on the metabolomic datasets, we fixed the random state to 0 for all six machine learning models. All metabolomic datasets were also normalized by the company.

### 2.2. Performance Metrics

To evaluate the effectiveness of the hybrid filter and embedded feature selection methods used for the metabolomic feature selection processes, we employed a task-specific metric: the classification accuracy rate for the classification tasks.

For classification, the accuracy rate (ACC) was calculated by dividing the sum of true positives and true negatives by the total number of samples (including false positives and false negatives) [15], as detailed in Equation (1).
(1)$$\mathrm{ACC}\;=\frac{\mathrm{TP}+\mathrm{TN}}{\mathrm{TP}+\mathrm{TN}+\mathrm{FP}+\mathrm{FN}}\;$$
where TP, FN, TN, and FP denote the number of true positives, false negatives, true negatives, and false positives, respectively.

### 2.3. Computational Results

This subsection presents the effect of our feature selection and weighting approach on the performance of the prediction model for three metabolomic datasets.

#### 2.3.1. The Effect of Feature Selection Methods on Classification Model Performance for Pre–Post-CRT, 12-Month OS, and 20-Month OS Status

Table 1, Table 2 and Table 3 present the average performance of six machine learning models on the metabolomic dataset for the pre–post-CRT, 12-month OS, and 20-month OS status classification, with and without feature selection using five-fold stratified cross-validation. The LASSO FS method gives the best result, with an 89.192% accuracy value and LR model when compared to the cases where only mRMR FS is used or FS methods are not used for the pre–post-CRT status classification. According to Table 2, the voting ensemble model without using the FS method gives the best result, with a 79.933% accuracy value for the 12 month-OS status-based dataset. As can be seen from Table 3, LASSO FS with the SVM model yields a 64.474% accuracy as the best result for the 20-month OS classification-based feature selection on the metabolomic dataset. These results underscore the significance of feature selection in enhancing model performance, especially for models such as LR and SVM. Both LASSO and mRMR offer effective methods for selecting features, but the optimal choice may vary based on the dataset’s characteristics and the specific model employed. Grasping how different feature selection techniques influence model performance is essential for creating robust predictive models. The changes in color, from red to green, in the tables represent the performance results, from the lowest (red) to the highest values (green).

#### 2.3.2. The Effect of Feature Selection and Weighting Method on Classification Model Performance for Pre–Post-CRT, 12-Month OS, and 20-Month OS Status

We assessed the performance of LASSO and mRMR feature selection methods using rank-based weighting schemes (i.e., weights: 1 and 2). The computational outcomes for these experiments, with ‘k’ and # representing the minimum weight count and number, are tabulated in detail in Table 4, Table 5 and Table 6. According to Table 4, the best possible result for the feature selection process for the pre–post-CRT metabolomic data is obtained with 48 features and a minimum weight of 6; this is using the soft voting of the five classification models employed, with a 96.711% accuracy value. The result also shows that LASSO = 2 and mRMR = 1 weights. For the 12-month OS status-based dataset (i.e., Table 5), the best result is obtained with 46 features and a minimum weight of 4; this is by using the LR model employed, with a 92.093% accuracy value and LASSO = 2 and mRMR = 1 weights. Additionally, according to Table 6, the soft voting of five machine learning models gives the highest accuracy rate of 86.910% by using 33 features and a minimum weight of 7. The names of the selected features for these three datasets are also illustrated in Table 7, Table 8 and Table 9.

To interpret these results theoretically, the situations can be summarized as follows: The soft voting ensemble model and LR consistently show high accuracy across various feature counts, indicating their effectiveness in feature selection. SVM also maintains a solid performance, particularly at higher feature counts. RF shows a stable performance, highlighting its robustness in handling feature variations. KNN generally shows the lowest initial performance and relatively minor improvements across feature counts. KNN does achieve high accuracy for some cases, such as an 87.364% accuracy with eight features for the pre–post-CRT status-based metabolomic dataset; but overall, it seems less affected by the number of features compared to other models. AdaBoost has a competitive performance, but it generally does not surpass LR. The soft voting ensemble method generally maintains solid accuracy, showcasing its strength in combining different models for improved predictions. As the number of features decreases, many models start to show diminishing returns in performance. For example, LR and SVM continue to achieve good accuracy even with fewer features, whereas KNN tends to struggle more when the feature count is low (i.e., more vulnerable to dimensionality issues).

### 2.4. Clinical Results

When employing the pre vs. post timepoints as class labels, 48 features were identified, 38 of which had biological annotations and 25 of which matched at least one existing database (Table 7). The feature name is also determined by combining the mass-to-charge ratio (*m*/*z*) value with the retention time (*rt*) of the underline. The top identified features were 138.0416_2.27 and 353.1606_10.7 (Figure 1A), resulting in an accuracy rate of 86% (Table 4); however, neither compound mapped to existing and known metabolomic databases such as KEGG for the interpretation operation. When employing 12-month OS as a class label, 46 features were identified, 36 of which had biological annotations and 23 of which matched at least one existing database (Table 8). The top two identified features were 132.9913_9.58 and 171.0409_2.39, which alone resulted in an 80% ACC (Table 5) (Figure 1B). When employing 20-month OS as a class label, 33 features were identified, 31 of which had biological annotations and 14 of which matched at least one existing database (Table 9). The identified compounds exhibited greater differences between classes with respect to 12-month OS compared to either pre vs. post-CRT or 20-month OS (Figure 2, Appendix A). Several compounds were shared between the pre vs. post-CRT analysis and 12-month OS (2 compounds) and between the pre vs. post-CRT and 20-month OS (3 compounds), while six compounds were shared between the 12-month OS and 20-month OS (Figure 3).

We also obtained the two most discriminative features (i.e., 138.0416_2.27 and 353.1606_10.7) with an 86.423% accuracy rate and the soft voting of five ML models (i.e., k = 14, LASSO = 2, and mRMR = 1) to enhance visualization by reducing the dimensionality for the pre–post-CRT status on the metabolomic dataset (see Figure 1). For the 12-month OS dataset, the two most discriminative features were 171.0409_2.39 and 132.9913_9.58. These features were selected with the LR model, with an 81.285% accuracy rate, and with a minimum weight of 11 by assigning weights to LASSO = 2 and mRMR = 1 (see Figure 1). The selection of only two features is also very useful in observing the value-based effects of the variables by providing the researchers with a better and more understandable visualization. It shows that the normalized value of these two features in Figure 1 can be easily separable by ML models. For Figure 1A, if the normalized 353.1606_10.7 compound value is greater than 0, it can be generally categorized as pre-CRT for all patients by using the 138.0416_2.27 compound as the second feature. Otherwise, the status can generally be classified as post-CRT for all patients in the metabolomic data. For Figure 1B, if the normalized 171.0409_2.39 compound value is less than 0, the OS status of all patients can be generally categorized as alive at 12 months by using the 132.9913_9.58 compound as the second feature. Otherwise, the status can generally be classified as dead at 12 months for all patients in the metabolomic data. The related figure indicates the difference in the compound levels between post-CRT and pre-CRT cases, those alive at 12 months and those dead at 12 months, and those alive at 20 months and those dead at 20 months; this is also given in Figure 2. The heatmap matrices for the mean values of the selected features with respect to the pre–post-CRT, 12-month OS, and 20-month OS status are also illustrated in Appendix A for the metabolomic dataset.

## 3. Discussion

This study explored the use of feature selection when applied to large-scale metabolomic data measured in the serum of patients with GBM prior to and after treatment with CRT, linking potential metabolomic changes that are capturable in serum to patients’ 12- and 20-month OS. This is the first study to employ serum metabolomic data acquired non-invasively at two time points in GBM. Several studies have employed metabolome data based on GBM cell lines [9], GBM tissue specimens [17,18] and biofluids (CSF [19], serum [20] and plasma [21]), and there is now considerable evidence that the metabolome may provide crucial insights into the biology of GBM, with altered metabolic signaling being a cornerstone of tumor resistance [22]. It appears that there are metabolic differences between GBM and healthy individuals [21] and between GBM and other gliomas [20]. Ferrasi et al., analyzed GBM plasma biomarkers in comparison with healthy individuals, identifying seven plasma biomarkers including arginylproline (*m*/*z* 294), 5-hydroxymethyluracil (*m*/*z* 143), and N-acylphosphatidylethanolamine (*m*/*z* 982) [21]. Moren et al., found that glioma subtypes harbor different metabolic feature patterns in tumor tissue and serum that are associated with outcomes, thus advancing the use of realistic biomarkers for the characterization of glioma [20]. However, since most data provide a snapshot of a single time point, we do not know how metabolic compounds of interest may behave when treatment is administered. In this analysis, several compounds that distinguish the serum signal pre vs. post-CRT in GBM and several others that relate to 12- and 20-month survival were identified. While 48 compounds were identified in relation to CRT, only 25 of these matched at least one commonly employed metabolomic database; for the majority, it is currently not possible to definitively determine their biological role. However, the emergence of compounds such as ondansetron, a commonly administered antiemetic during CRT, suggests the potential clinical validity of the identified features. The top two identified compounds in relation to CRT were 138.0416_2.27 and 353.1606_10.7, and independently of the remaining 46 compounds, these two were able to distinguish the pre vs. post signal with 86% accuracy. However, they are challenging to untangle clinically since 138.0416_2.27 is not biologically annotated and 353.1606_10.7 maps to Inulicin with level 3 confidence. A further analysis of 353.1606_10.7 involving its spectrum revealed that it is more likely to represent Iclaprim, C_19_H_22_N_4_O_3_ or 3-(4-(2-Dimethylamino-1-methylethoxy)phenyl)-1H-pyrazolo(3,4-b)pyridine-1-acetic acid and C_19_H_22_N_4_O_3_ (Figure 4). The precise role of either compound in relation to the alteration in serum pre vs. post-CRT in GBM thus remains currently unclear.

With respect to the compounds identified in connection to 12-month survival, 132.9913_9.58 (which mapped to 2-Furoate with confidence level 3) and 171.0409_2.39 (which mapped to Thymine with confidence level 3) could distinguish between patients who were alive at 12 months and those who had died by 12 months following diagnosis, with 80% accuracy (Figure 5). Both compounds were generally elevated in patients who had died within 12 months of diagnosis compared to those patients that lived more than 12 months, although overlap was also noted given the 80% accuracy rate. The lack of robust biological annotation here again makes clinical connections difficult. We do note that 6-methyl uracil represents a possible compound for the unknown 171.0409_2.39, which had mapped to thymine with confidence level 3 (Figure 5). We also note that 5-hydroxymethyluracil has been reported as differentially altered between GBM and healthy individuals in plasma [21]. The elevation of uracil mediators observed in patients with poorer outcomes could indicate a possible connection to the amino acid and carbon metabolism in GBM [23]. The amino acid and carbon metabolism may be associated with the energy currency in GBM as a means of tumor resistance, leading to inferior survival or enhanced DNA repair [24]. The identification of uracil as the compound in this analysis also links it to proteomic markers we have previously identified in connection to the status and expression of MGMT with respect to MTHFR [4] and the unfolded protein response in proteomic alterations in GBM [25]. Ferrasi et al., in their plasma metabolome analysis, identified plasma pyruvate as an important discriminating biomarker but appropriately noted that pyruvate is not specific to GBM and hence evaluated its association with other biomarkers including 5-hydroxymethyluracil, proposing this association as useful in monitoring GBM [21]. This highlights the importance of systematically evaluating the associations of significant features against other discriminating biomarkers identified in the literature. We did identify several compounds with the level of confidence 1 [16], including in association with CRT: alanine/sarcosine guanine, dimer of chenodeoxycholyglycine (GCDCA). In connection with survival, 2-hydroxyhippuric acid (a glycine derivative), pantothenic acid (vitamin B5, a component of coenzyme CoA) and caffeine were identified for 12-month OS and L-threonine+Na was identified for 20-month OS. These compounds, which are present in biological systems (compared to the other identified compounds in this analysis), are well defined in terms of their significance within metabolic pathways, notably in the amino acid metabolism. A previous analysis of the metabolome, comparing tissue and serum expression, suggested that certain metabolic compounds are associated with poorer outcomes in both high and low-grade tumors, although unfortunately none of the serum compounds associated with GBM could be identified [20]. Serum alanine and threonine were compounds identified in a previous analysis, with an increased level associated with improved outcomes for oligodendroglioma. Of note, in this as well as in other analyses that involve the metabolome and serum sample analysis, amino acids feature most prominently as significant signals, indicating both capture in serum as well as a connection to tumor biology (grade) and outcomes, thus rendering amino acid compounds a promising avenue to watch when examining the metabolome in glioma. Alanine and threonine are also serum metabolites associated with an increased risk of developing high-grade glioma in a prospective analysis, as is the caffeine metabolism [26], potentially linking this to our identification of caffeine as a metabolite associated with 12-month OS. The dimer of chenodeoxycholyglycine (GCDCA), also referred to as glycochenodeoxycholic acid, is a bile acid derivative that plays a key role in metabolic processes and is particularly critical given its association with mitochondrial dysfunction [27], its ability to cross the blood–brain barrier, and its connection to individuals’ response to CRT, although its implications in GBM require further study [28]. In a recent publication examining the plasma levels of biogenic amines in GBM, surgery was associated with increased levels of glycodeoxycholic acid and several amino acids; meanwhile, CRT was associated with altered levels of six other metabolites including several glycine derivatives with a distinct metabolic profile pre vs. post-surgery and pre vs. post-CRT [29]. The levels of amino acids, notably uracil and bile-acid-pathway-related compounds, can have an effect on tumor behavior and reprogramming, resulting in altered proliferation, viability, and migration, as has been described in previous research involving GBM cell lines, with significant linkages to solute carriers and large amino acid transporters [30]. Future directions include the analysis of shared features of the metabolome and the proteome in conjunction with the clinical features and molecular classification of specific compounds with robust biological annotation. Ongoing research will be needed to annotate, link, and validate signals to determine how the metabolism is associated with tumor resistance and outcomes in GBM. This study illustrates, however, that the serum metabolism can provide a promising, noninvasive avenue for the future study of this devastating disease. The classification models’ parameters can be appropriately tuned and optimized to achieve optimal performance and obtain new and interesting results by taking into account the computational load and time requirements. As another future direction of this work, we can also handle imaging data such as MRI using AI-driven methodologies (e.g., classification or clustering [31]) for biomarker discovery for GBM-related tasks.

The limitations of this study include the retrospective nature of the study and the extended time period over which the patients included in the analysis were treated. From a data standpoint, limitations include the limited biological annotation of metabolome compounds at this time given the ongoing evolution of large-scale data and the database matching of measured compounds with linkage to biologic pathways. Of the total amount of compounds that emerged across the three analyses, only seven compounds carried a confidence level of 1 and there is also the possibility that biologically identified compounds with identification based on the mass-to-charge ratio combined with retention time may present inaccuracies given that a mass-to-charge ratio can be associated with several compounds. In addition, alterations in the measurements of compounds pre vs. post-CRT are small in magnitude compared with what may possibly be observed in tissue-based gene expression studies, with the tradeoff being that this modality of measurement is non-invasive. An added limitation is the extended time period between sample acquisition and the survival time points employed for the classification of 12 and 20 months. We elected to use 12 months since this typically marks the completion of adjuvant TMZ, which is typically administered for 6 to 12 cycles, and reported survival based on historical data [32]. This is also supported by survival data for GBM in real-world settings [33] and, pragmatically, the closest time point to sample acquisition where early progressors may be metabolically distinguished from patients with more sustained responses. The 20-month OS timepoints are backed up by data spanning several trials as the current median OS time point [32].

## 4. Materials and Methods

This section provides an overview of the dataset employed (Department of Pathology, Immunology and Laboratory Medicine University of Florida, Gainesville, FL, USA) and the key characteristics of our metabolomic datasets. Subsequent subsections detail our methodological approach, including the definitions, techniques, and prediction models utilized in this study.

### 4.1. Datasets

We employed three metabolomic datasets to select feature sets for the classification tasks in this study. The metabolomic dataset consists of 107 patients diagnosed with pathology-proven GBM between 2005 and 2023. All patients underwent upfront CRT. Serum samples were collected prior to the initiation of CRT (average of 6.7 days, ranging from 0 to 24 days) and after its completion (average of 0.33 days, ranging from −1 to 31 days) [4,25]. The average interval between pre- and post-sample acquisition was 48, with a range of 22 to 83 days [4]. Following collection, the serum samples were stored at −80 °C for an average of 3951 days (ranging from 239 to 7072 days) [34]. The characteristics of our metabolomic dataset are given in Table 10. The dataset-storing operations were performed by the NIDAP environment [35].

### 4.2. Methodology

We provide a general framework of our utilized method in this section, including a description of the feature selection and weighting techniques, and a brief overview of the classification models used.

#### 4.2.1. Proposed Scheme

In this study, we utilize a hybrid approach for feature weighting and selection approaches based on rank [19], aimed at categorizing the pre–post-CRT, 12-month, and 20-month OS statuses in metabolomic data. Our methodology comprises two main phases: (i) feature selection (FS) and (ii) feature weighting (FW) [4,5]. Figure 6 illustrates an algorithmic diagram of our architecture, highlighting the two feature selection methods employed: Least Absolute Shrinkage and Selection Operator (LASSO) and Minimum Redundance Maximum Relevance (mRMR).

Initially, all metabolomic features are input into the feature selection (FS) model using a cross-validation technique. For each fold, the feature sets chosen by the two FS methods are recorded, and their counts are adjusted according to the weights assigned by the rank-based approach [4]. Subsequently, the minimum weight-based feature list is assessed with all weight values. In the final stage, we derive the final selected feature list by evaluating all weight values and identifying those with the highest accuracy rate via six machine learning models (i.e., Support Vector Machine (SVM), Logistic Regression (LR), K Nearest Neighbors (KNN), Random Forest (RF), Adaptive Boosting (AdaBoost), and Voting). The main principles of the LASSO and mRMR feature selection methods are briefly described as follows:

LASSO is a type of analysis method that performs feature selection and regularization to enhance the prediction performance and interpretability of the statistical models. It applies the L1 norm penalty to the coefficients by shrinking some of the coefficients to exactly zero, effectively performing feature selection by excluding irrelevant features (i.e., sparsity). The selected features are those that contribute most to the learning model, reducing dimensionality and mitigating overfitting. The mRMR (Minimum Redundancy Maximum Relevance) method is a filter-based feature selection technique that aims to identify the features most pertinent to the target variable while reducing redundancy (e.g., correlated) among the selected features. The mRMR method ensures that the selected features not only predict the targets well but also complement each other by avoiding redundant information.

The hybridization of LASSO (Least Absolute Shrinkage and Selection Operator) and mRMR (Minimum Redundancy Maximum Relevance) feature selection methods works effectively as it leverages the complementary strengths (i.e., regularization and sparsity for LASSO, and balancing feature redundancy and relevance for mRMR) of both techniques to improve the prediction performance and provide a more robust feature selection process and ensure relevant feature diversity by addressing the limitations of each feature selection method separately. A detailed description of this approach can be obtained from Tasci et al. [4].

#### 4.2.2. Feature Selection Methods

Feature selection is a technique used to streamline data by identifying and retaining only the most relevant features. By eliminating redundant, irrelevant, or noisy information, this process enhances the model’s performance, interpretability, and efficiency [8,12]. Addressing the challenges posed by high-dimensional datasets, feature selection helps overcome computational constraints, improves data visualization, and bolsters model accuracy in real-world applications. The commonly employed methods can be categorized into three primary groups: filter, wrapper, and embedded approaches; each differs in how feature subsets are evaluated [12]. Filter methods assess features individually without considering the model, using metrics like statistical measure (e.g., mRMR). Wrapper methods evaluate feature subsets by repeatedly training and testing models, optimizing feature selection for specific models. Embedded methods integrate feature selection within the model-building process itself (e.g., LASSO regularization). The LASSO and mRMR FS methods adopted in this paper were applied and weighted according to their performances, as detailed in Tasci et al. [4].

#### 4.2.3. Feature Weighting Methods

To assess the relative importance of selected features in distinguishing patterns, we assigned weights based on their contribution (i.e., accuracy rate) to the classification operation. In this study, we employed a rank-based weighting scheme for the LASSO and mRMR FS methods. They were ranked by their performance (accuracy) on each data fold. The top-performing method received a weight of 2, while the other received a weight of 1. This process was repeated for all possible rank combinations to optimize the feature weights for metabolomic datasets.

### 4.3. Classification

Classification is a core machine learning task that assigns predefined labels to data based on their characteristics. The aim is to create a model capable of accurately categorizing new, unseen data. The machine learning models are trained on labeled datasets to learn underlying patterns, enabling them to make predictions. Common classification algorithms include KNN, SVM, LR, AdaBoost, and RF, which are detailed in our previous work [4]. In addition to these classifiers, we also applied ensemble soft voting to these five prediction models as a new machine learning model in this study. Unlike hard voting, which combines class labels from multiple classifiers, soft voting utilizes probability scores [36,37]. The final prediction is determined by combining these probabilities using a specific method, such as the maximum of the sums of the predicted probabilities [38].

## 5. Conclusions

This study emphasizes the essential role of feature selection in the analysis of metabolomic data, showing that effective selection techniques can greatly improve model performance and clarity. By pinpointing the most relevant compounds linked to specific biological conditions, we can enhance our understanding of metabolic pathways and disease mechanisms. The proposed method uniquely addresses feature selection and weighting using large-scale metabolomic data measured in the serum of patients with GBM at two-time points pre and post-SOC CRT. Several promising compounds were identified, with the ability to distinguish signal pre-CRT vs. post-CRT with 96.711% accuracy, 12-month overall survival with 92.093% accuracy and 20-month overall survival with 86.910% accuracy. The high-level biological annotation of growing metabolomic data, including mapping to signaling pathways, is still evolving. These data are less mature compared to other omics data types.

Our findings reveal that hybrid feature selection methods surpassed traditional techniques, resulting in improved classification accuracy and more reliable predictive models. This highlights the necessity of using advanced methods that are suited to the complexities of metabolomic data.

Looking ahead, we advocate for the integration of feature selection processes into standard metabolomic workflows to support biomarker discovery and promote advancements in personalized medicine.

## Figures and Tables

**Figure 1 ijms-25-10965-f001:**
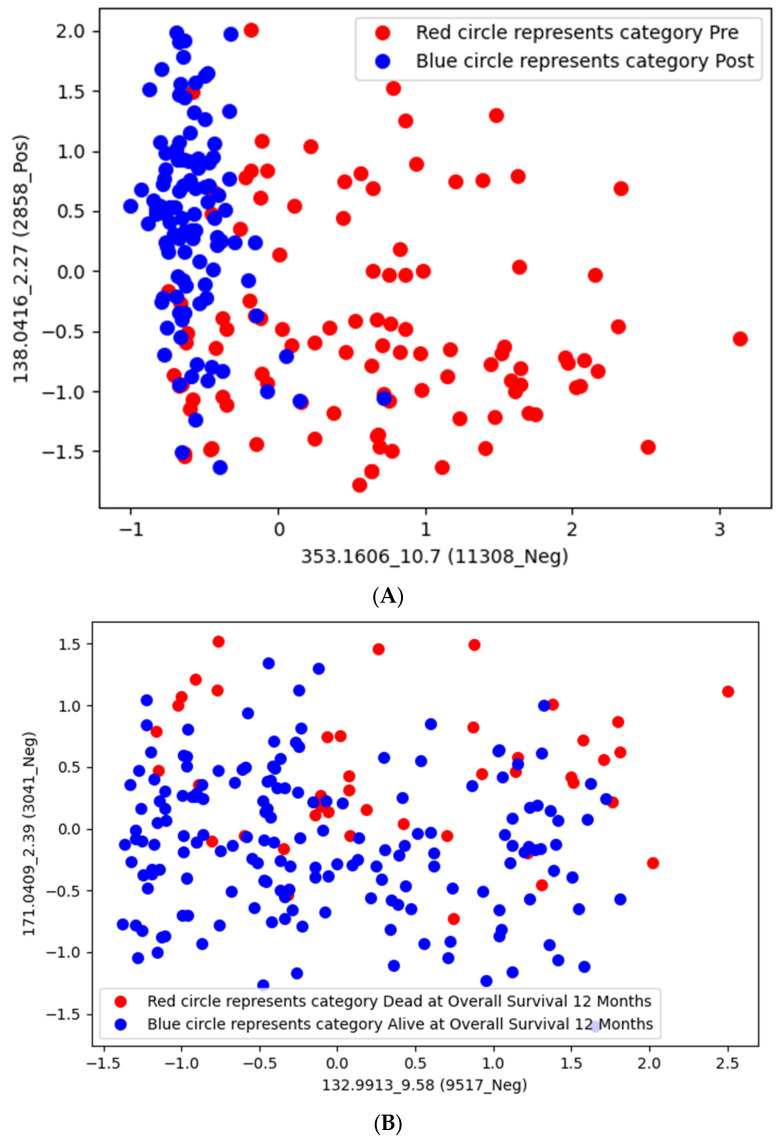
(**A**) Two of the most significant features that classify the pre–post-CRT-based statuses for the metabolomic dataset. (**B**) The two most significant features that classify the 12-month OS-based statuses for the metabolomic dataset.

**Figure 2 ijms-25-10965-f002:**
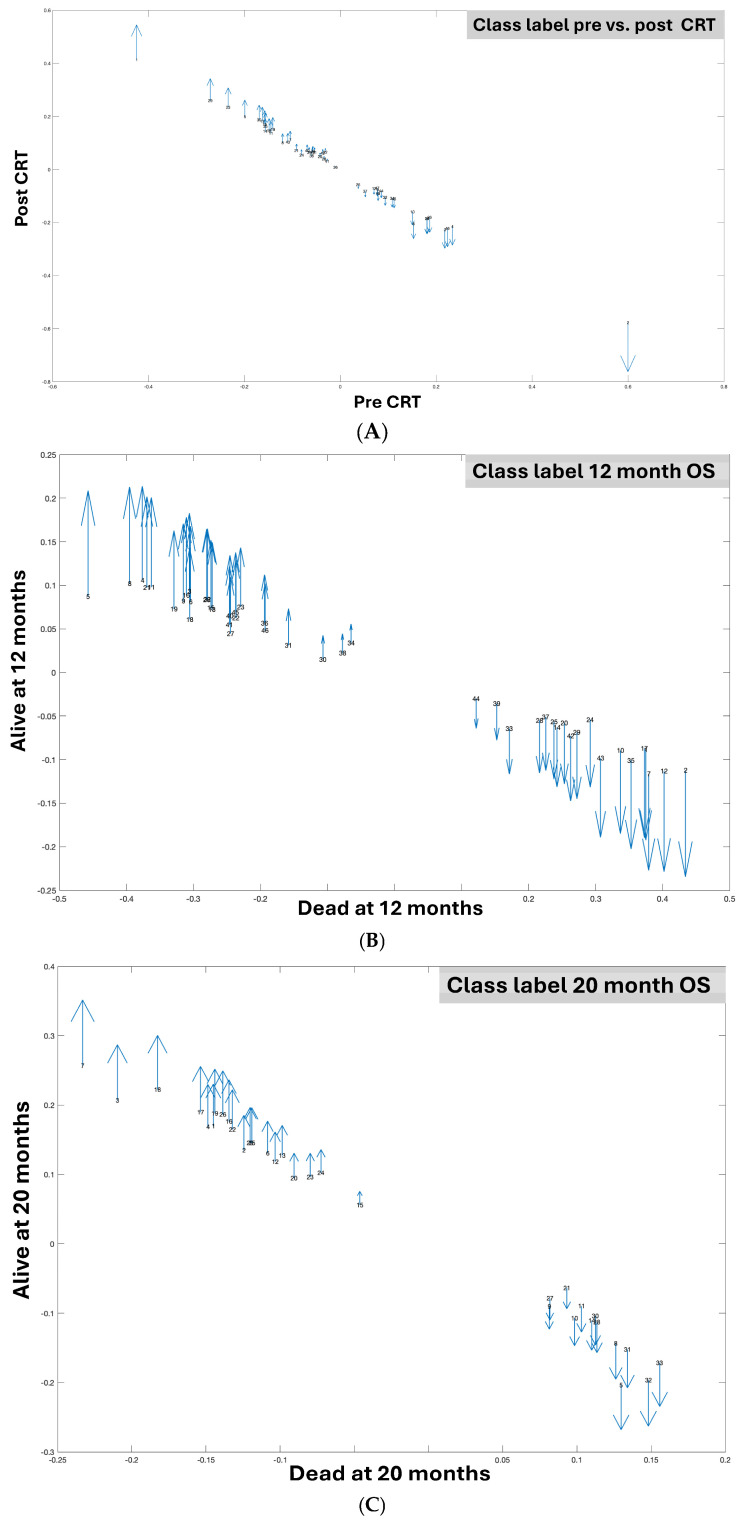
The features identified in the local dataset are associated with (**A**) the pre–post-CRT status; (**B**) 12-Month OS; and (**C**) 20-Month OS. Blue arrows indicate the difference in compound levels between post-CRT and pre-CRT cases, those alive at 12 months and those dead at 12 months, and those alive at 20 months and those dead at 20 months (up arrow = higher, down arrow = lower).

**Figure 3 ijms-25-10965-f003:**
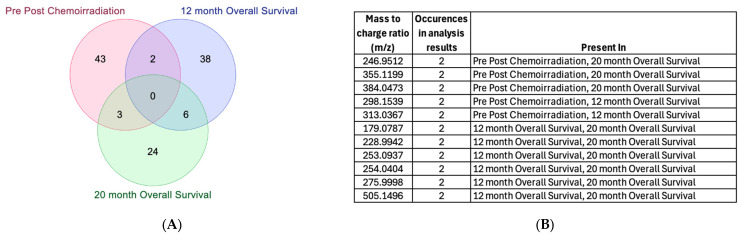
(**A**) Number of features shared between pre–post-CRT, 12-month overall survival, and 20-month overall survival analysis. (**B**) The shared metabolomic features identified and expressed by the mass-to-charge ratio (*m*/*z*). The biological annotation of the identified featured is currently lacking.

**Figure 4 ijms-25-10965-f004:**
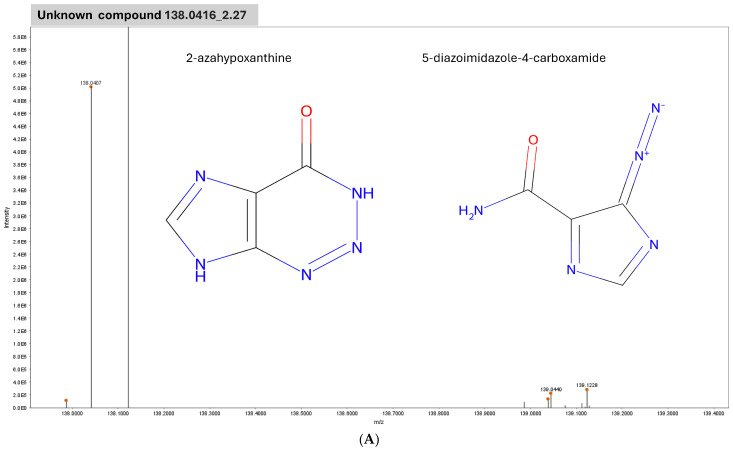
Isotopic pattern and chemical structures of the two compounds identified with pre/post classification with 86% ACC. (**A**) 138.0416_2.27 (no biological annotation) and (**B**) 353.1606_10.7 (which mapped to Inulicin with confidence level 3), which, per subsequent analysis, is most likely to represent Iclaprim, C_19_H_22_N_4_O_3_ or 3-(4-(2-Dimethylamino-1-methylethoxy)phenyl)-1H-pyrazolo(3,4-b)pyridine-1-acetic acid, C_19_H_22_N_4_O_3_.

**Figure 5 ijms-25-10965-f005:**
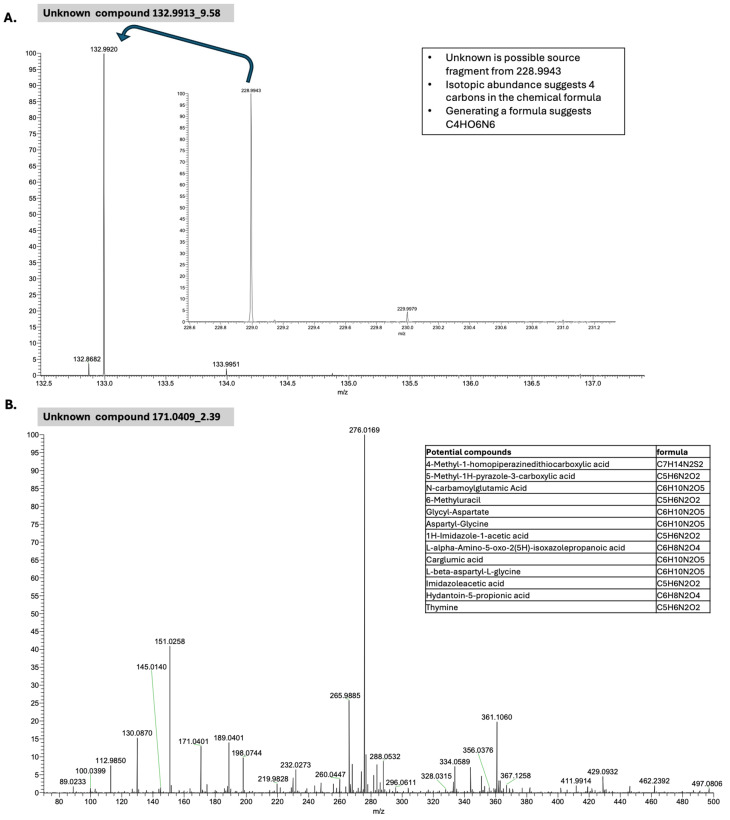
Isotopic pattern and chemical structures of the two compounds identified with 12-month alive/dead classification with 80% ACC. (**A**) 132.9913_9.58 (which mapped to 2-Furoate with confidence level 3). The bolded blue arrow indicates that unknown compound 132.9913_9.58 may represent a possible source fragment from 228.9943. (**B**) 171.0409_2.39 (which mapped to Thymine with confidence level 3), which, per subsequent analysis, may represent several compounds, as described in the side panels.

**Figure 6 ijms-25-10965-f006:**
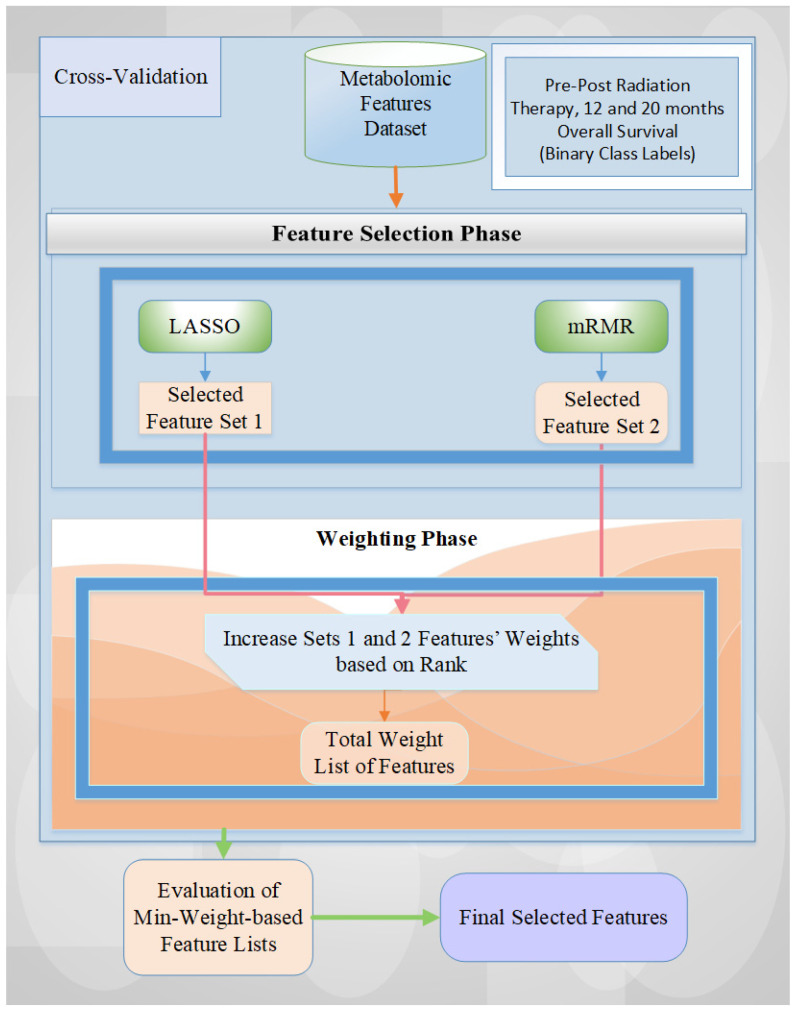
The general overview of the architecture utilized for metabolomic-based feature selection tasks. Red and green arrows show the associated and the following processes.

**Table 1 ijms-25-10965-t001:** The impacts of feature selection methods on the accuracy rate (%) for 214 of the 107 patients (i.e., pre- and post-CRT) according to the pre–post-CRT status-based metabolomic dataset.

ML-ACC	Before FS	LASSO	mRMR
**SVM**	70.975	86.866	86.888
**LR**	77.043	89.192	87.829
**KNN**	58.904	79.380	86.899
**RF**	70.986	88.317	86.434
**AdaBoost**	85.471	88.306	84.097
**Vote**	75.161	86.877	86.434

**Table 2 ijms-25-10965-t002:** The impacts of feature selection methods on the accuracy rate (%) for 214 of the 107 patients (i.e., pre- and post-CRT) according to the 12-month overall survival status-based metabolomic dataset.

ML-ACC	Before FS	LASSO	mRMR
**SVM**	79.446	76.645	71.938
**LR**	77.077	77.121	76.146
**KNN**	78.051	78.981	72.436
**RF**	78.516	78.970	76.179
**AdaBoost**	75.249	73.865	71.971
**Vote**	79.933	78.516	75.216

**Table 3 ijms-25-10965-t003:** The impacts of feature selection methods on accuracy rate (%) for 214 cases of the 107 patients (i.e., pre- and post-CRT) according to the 20-month overall survival status-based metabolomic dataset.

ML-ACC	Before FS	LASSO	mRMR
**SVM**	59.812	64.474	55.149
**LR**	58.372	60.709	57.951
**KNN**	53.289	61.661	47.663
**RF**	48.162	60.277	51.860
**AdaBoost**	46.312	53.256	44.806
**Vote**	60.221	63.057	53.289

**Table 4 ijms-25-10965-t004:** The impact of the rank-based feature weighting and selection method on the accuracy rate (%) for 214 of the 107 patients (i.e., pre- and post-CRT) according to the pre–post-CRT status-based metabolomic dataset.

LASSO = 1 and mRMR = 2							
k	# of Features	SVM	LR	KNN	RF	AdaBoost	Vote
1	252	92.934	95.282	80.310	91.119	87.830	93.389
2	116	93.898	95.781	84.098	92.990	89.247	94.385
3	55	95.293	95.305	85.039	91.595	91.108	96.235
4	33	95.293	95.304	88.328	93.931	89.712	95.758
5	23	95.781	94.861	86.456	90.631	89.723	93.920
6	11	91.561	91.561	89.236	89.701	87.342	91.096
7	9	90.166	91.096	87.841	90.188	85.925	90.166
8	6	89.701	89.247	88.749	89.236	83.145	88.760
9	6	89.701	89.247	88.749	89.236	83.145	88.760
10	4	84.109	85.039	82.669	87.364	79.889	84.097
11	4	84.109	85.039	82.669	87.364	79.889	84.097
12	4	84.109	85.039	82.669	87.364	79.889	84.097
13	3	85.991	84.596	85.969	87.852	78.959	87.852
14	3	85.991	84.596	85.969	87.852	78.959	87.852
**LASSO = 2 and mRMR = 1**							
**k**	**# of Features**	**SVM**	**LR**	**KNN**	**RF**	**AdaBoost**	**Vote**
1	252	92.934	95.282	80.310	93.433	87.830	93.389
2	230	92.934	95.747	81.705	90.642	89.712	93.853
3	92	94.839	95.770	83.621	90.631	88.284	95.304
4	89	94.374	95.770	84.551	92.514	90.665	95.304
5	49	93.887	95.781	83.189	91.617	91.119	96.246
6	**48**	93.887	96.246	84.119	93.466	89.258	**96.711**
7	27	94.850	94.850	85.061	90.166	92.049	96.257
8	25	93.909	94.839	85.526	91.572	89.712	94.396
9	18	92.979	92.492	85.958	90.642	90.166	93.433
10	17	91.561	92.027	85.947	91.584	88.749	92.957
11	8	87.353	88.284	87.364	89.712	83.145	89.236
12	5	86.899	86.434	84.562	86.899	79.889	86.423
13	3	85.991	84.596	85.969	86.910	78.959	87.841
14	2	84.109	82.248	85.969	84.563	78.937	86.423

**Table 5 ijms-25-10965-t005:** The impact of the rank-based feature weighting and selection method on the accuracy rate (%) for 214 of the 107 patients (i.e., pre- and post-CRT) according to the 12-month overall survival status-based metabolomic dataset.

LASSO = 1 and mRMR = 2							
k	# of Features	SVM	LR	KNN	RF	AdaBoost	Vote
1	191	85.072	89.745	80.376	78.516	79.911	89.291
2	76	87.896	89.756	81.296	80.842	81.340	89.767
3	38	85.526	88.339	82.248	80.399	79.003	86.944
4	25	88.782	90.665	80.853	81.329	81.340	87.386
5	16	81.761	85.050	79.435	81.307	78.992	83.156
6	14	80.853	84.130	80.365	79.934	82.747	81.296
7	9	82.248	83.167	80.820	82.259	79.435	82.713
8	4	79.922	81.783	78.062	76.667	79.469	80.853
9	1	79.457	79.922	78.505	71.982	78.992	78.051
10	1	79.457	79.922	78.505	71.982	78.992	78.051
**LASSO = 2 and mRMR = 1**							
**k**	**# of Features**	**SVM**	**LR**	**KNN**	**RF**	**AdaBoost**	**Vote**
1	191	85.072	89.745	80.376	78.981	79.911	89.291
2	172	84.607	89.280	80.852	79.446	78.981	89.291
3	55	87.896	89.302	80.365	79.911	80.853	87.442
4	**46**	89.291	**92.093**	80.376	80.841	83.654	90.233
5	25	86.456	91.141	81.772	80.376	81.329	87.398
6	21	86.933	87.874	83.167	81.794	81.329	86.456
7	15	86.002	85.991	82.702	81.329	79.933	85.537
8	15	86.002	85.991	82.702	81.329	79.933	85.537
9	8	80.376	85.504	80.831	81.307	80.809	82.237
10	5	78.029	81.750	78.007	79.900	81.772	80.354
11	2	78.494	81.285	77.563	78.948	75.714	79.878

**Table 6 ijms-25-10965-t006:** The impact of the rank-based feature weighting and selection method on the accuracy rate (%) for 214 of the 107 patients (i.e., pre- and post-CRT) according to the 20-month overall survival status-based metabolomic dataset.

LASSO = 1 and mRMR = 2							
k	# of Features	SVM	LR	KNN	RF	AdaBoost	Vote
1	364	71.528	71.451	59.812	64.053	51.406	69.579
2	162	79.457	76.102	64.928	70.554	58.450	81.274
3	70	82.226	81.285	69.624	70.067	64.020	79.889
4	41	83.200	83.189	71.030	74.308	69.181	80.376
5	23	77.619	79.479	69.192	72.492	62.614	79.490
6	14	74.319	76.157	71.949	74.795	63.544	76.667
7	9	74.784	72.868	68.716	67.763	62.558	73.367
8	6	64.950	66.346	55.592	58.837	56.478	63.533
9	6	64.950	66.346	55.592	58.837	56.478	63.533
10	3	59.779	63.045	56.999	56.556	51.428	57.929
11	2	52.303	60.698	47.674	48.627	45.349	48.106
12	1	57.464	59.336	55.149	50.930	47.220	54.662
**LASSO = 2 and mRMR = 1**							
**k**	**# of Features**	**SVM**	**LR**	**KNN**	**RF**	**AdaBoost**	**Vote**
1	364	71.528	71.451	59.812	57.940	53.787	71.462
2	339	72.447	73.787	56.545	60.288	55.593	71.440
3	137	80.819	79.845	58.904	64.961	58.793	79.867
4	132	80.831	81.241	58.870	70.576	64.463	79.391
5	65	84.563	84.540	71.041	74.275	67.298	80.365
6	59	85.482	85.471	68.250	73.356	74.817	82.215
7	**33**	84.585	83.655	73.355	72.891	64.950	**86.910**
8	31	81.329	83.189	70.122	71.008	67.752	82.724
9	18	74.828	77.608	69.214	67.807	63.953	74.352
10	15	74.374	75.260	61.274	65.482	62.093	72.923
11	8	70.565	72.902	62.182	63.533	59.269	67.763
12	5	64.020	65.415	57.486	57.962	51.362	59.801
13	1	50.410	59.247	45.327	43.931	40.675	46.257

**Table 7 ijms-25-10965-t007:** This table shows the 48 identified features associated with the administration of concurrent chemoirradiation (CRT). The top two features, 138.0416_2.27 and 353.1606_10.7, alone resulted in an 86% ACC. *m*/*z* represents the mass-to-charge ratio, and rt represents the retention time. The biological annotation and its associated levels of confidence are listed, where 1 is the highest level matching to authentic standards (in bold), 2 is putatively annotated with the annotation verified using additional techniques, 3 is putatively identified using the compound class and 4, or NaN (network of advanced NMR), is unknown [16].

Rank of Importance	Polarity	*m*/*z*	*rt*	Possible Biological Annotation	Confidence Level
1	+	138.0416	2.27	138.0416_2.27	unknown
2	−	353.1606	10.70	Inulicin	3
3	−	180.1029	11.70	Tussilagine	3
4	+	224.1122	0.74	Mannitol	3
5	−	257.0780	2.91	(5-L-Glutamyl)-L-glutamine	3
6	−	246.9512	9.53	Bropirimine	3
7	−	264.8304	0.97	264.8304_0.97	unknown
8	+	90.0553	0.83	ALANINE/SARCOSINE	**1**
9	−	313.0367	9.53	313.0367_9.53	unknown
10	+	138.0424	6.26	138.0424_6.26	unknown
11	+	355.1199	10.09	Timolol	3
12	+	436.2316	12.11	Nigakilactone M	3
13	+	287.8872	7.76	287.8872_7.76	unknown
14	+	214.0877	7.31	Trihomomethionine	3
15	−	305.1505	8.87	Baptifoline	3
16	−	356.0989	7.20	Taxiphyllin	3
17	+	122.0582	5.21	N-Methyl-2-pyrrolidinone	3
18	−	300.0187	7.92	2-(3,5-Dichlorophenylcarbamoyl)-1,2-dimethylcyclopropane-1-carboxylic acid	3
19	+	258.0914	6.91	Oxaprozin	3
20	+	425.1692	7.17	Norbixin	3
21	+	361.1822	8.09	Zwittermicin A	3
22	+	287.6615	7.91	287.6615_7.91	unknown
23	+	295.1642	9.75	Methyl farnesoate	3
24	+	354.0709	10.20	Malvidin	3
25	+	298.1539	9.44	GA	3
26	−	296.9815	9.58	Menadione sodium bisulfite	3
27	−	623.2873	9.01	Taxine A	3
28	−	446.3772	13.79	446.3772_13.79	unknown
29	+	294.1590	9.75	Ondansetron	3
30	−	298.0883	7.90	7-Mercaptoheptanoylthreonine	3
31	+	370.2937	13.58	370.2937_13.58	unknown
32	−	592.3527	13.61	Zizyphine A	3
33	−	298.1416	7.32	Promacyl	3
34	−	827.5642	16.23	827.5642_16.23	unknown
35	−	331.0828	9.80	Sulochrin	3
36	−	268.0408	9.39	Nifurtimox	3
37	−	384.0473	9.76	1-(5-Phospho-D-ribosyl)-5-amino-4-imidazolecarboxylate	3
38	+	212.1178	8.82	Trp-P-1	3
39	+	152.0555	1.37	GUANINE	**1**
40	−	897.6224	11.40	DIMER OF CHENODEOXYCHOLYGLYCINE (GCDCA)	**1**
41	−	246.9918	8.61	4-Sulfobenzoate	3
42	−	309.0748	12.02	Dikegulac	3
43	−	121.0659	9.08	Phenylethyl alcohol	3
44	−	193.0898	10.10	193.0898_10.1	unknown
45	+	233.1275	7.07	N-Caffeoylputrescine	3
46	−	243.1714	8.20	N-(6-Aminohexanoyl)-6-aminohexanoate	3
47	−	281.0105	9.84	Pseudopurpurin	3
48	−	293.0164	7.75	unsym-Bis(4’-chlorophenyl)ethylene	3

**Table 8 ijms-25-10965-t008:** The 46 identified features associated with 12-month overall survival. The top two features, 132.9913_9.58 and 171.0409_2.39, alone resulted in an 80% ACC. The biological annotation and its associated levels of confidence are listed, where 1 is the highest level matching authentic standards (in bold), 2 is putatively annotated, with the annotation verified using additional techniques, 3 is putatively identified using the compound class and 4, or NaN (Network of advanced NMR), is unknown [16].

Rank of Importance	Polarity	*m*/*z*	*rt*	Possible Biological Annotation	Confidence Level
1	−	132.9913	9.58	2-Furoate	3
2	−	171.0409	2.39	Thymine	3
3	+	453.7457	7.91	453.7457_7.91	unknown
4	−	505.1496	10.86	Paucin	3
5	−	226.0181	9.20	Penicillenic acid	3
6	+	253.0937	9.79	Nebularine	3
7	+	196.0599	7.39	N-Acetyl-L-2-amino-6-oxopimelate	3
8	+	179.0787	1.23	Methylenediurea	3
9	−	313.0367	9.53	313.0367_9.53	unknown
10	−	293.9709	7.41	Dichlozoline	3
11	−	359.0985	7.56	Glucovanillin	3
12	−	194.0458	7.36	2-HYDROXYHIPPURIC ACID	**1**
13	−	185.0023	10.90	185.0023_10.9	unknown
14	+	244.0705	10.02	Guanidinoethyl methyl phosphate	3
15	+	460.7339	8.90	460.7339_8.9	unknown
16	−	254.0404	9.88	Carprofen	3
17	−	228.9942	9.57	DCEBIO	3
18	−	199.8050	8.55	199.805_8.55	unknown
19	−	453.2163	9.91	Tetrahydrogeranylgeranyl diphosphate	3
20	−	218.1035	6.74	PANTOTHENIC ACID	**1**
21	+	188.1024	0.86	(-)-Hygroline	3
22	+	194.6184	2.31	194.6184_2.31	unknown
23	+	291.1444	8.35	Mepivacaine	3
24	−	225.0771	10.53	Genipin	3
25	−	199.0977	9.66	O-Propanoylcarnitine	3
26	−	356.1178	9.82	1-Carbazol-9-yl-3-(3,5-dimethylpyrazol-1-yl)-propan-2-ol	3
27	−	394.9228	8.60	394.9228_8.6	unknown
28	−	160.0616	1.37	Streptamine	3
29	−	275.9998	7.93	Pyrifenox	3
30	+	298.1539	9.44	GA	3
31	−	240.9815	8.27	4-Chloro-4′-biphenylol	3
32	−	206.8544	0.98	Tetrathionate	3
33	−	319.1664	9.72	Allyxycarb	3
34	−	176.0387	1.16	N-Formylmethionine	3
35	−	209.0570	6.77	Deoxycytidine	3
36	−	227.0387	10.10	Chloromethiuron	3
37	−	157.0508	8.66	Calystegin B2	3
38	−	245.8171	0.96	Pentachlorophenol	3
39	−	411.1290	9.17	1alpha,5alpha-Epidithio-17a-oxa-D-homoandrostan-3,17-dione	3
40	+	388.2196	7.93	L-Alanyl-gamma-D-glutamyl-L-lysine	3
41	−	495.2237	9.55	Nigakilactone E	3
42	−	224.0235	1.23	S-Carboxymethyl-L-cysteine	3
43	+	195.0874	8.00	CAFFEINE	**1**
44	−	434.2196	12.07	434.2196_12.07	unknown
45	+	749.3806	8.48	749.3806_8.48	unknown
46	+	146.1521	7.56	146.1521_7.56	unknown

**Table 9 ijms-25-10965-t009:** The 33 identified features associated with 20-month overall survival. The biological annotation and its associated levels of confidence are listed, where 1 is the highest level matching authentic standards (in bold), 2 is putatively annotated, with the annotation verified using additional techniques, 3 is putatively identified using the compound class and 4, or NaN (Network of advanced NMR), is unknown [16].

Rank of Importance	Polarity	*m*/*z*	*rt*	Possible Biological Annotation	Confidence Level
1	+	253.0937	9.79	Nebularine	3
2	−	279.1254	10.28	Lycodine	3
3	−	384.0473	9.76	1-(5-Phospho-D-ribosyl)-5-amino-4-imidazolecarboxylate	3
4	+	179.0787	1.23	Methylenediurea	3
5	−	246.9512	9.53	Bropirimine	3
6	+	142.0475	0.87	L-THREONINE+Na	**1**
7	+	355.1199	10.09	Timolol	3
8	−	627.3754	10.60	627.3754_10.6	unknown
9	−	275.9998	7.93	Pyrifenox	3
10	+	207.1099	7.08	3-Dehydrocarnitine	3
11	−	230.9968	8.72	4-Sulfobenzaldehyde	3
12	−	348.0705	6.47	Atovaquone	3
13	−	204.9466	12.12	3,5-Dichloro-2-methylmuconate	3
14	+	184.0828	9.93	N(omega)-Nitro-L-arginine	3
15	−	323.0885	11.51	Portulacaxanthin I	3
16	+	160.0869	9.88	Benzimidazole	3
17	−	505.1496	10.86	Paucin	3
18	−	254.0404	9.88	Carprofen	3
19	−	371.1899	10.47	Emopamil	3
20	−	220.9842	0.71	D-Erythrose 4-phosphate	3
21	−	571.3001	7.91	571.3001_7.91	unknown
22	−	503.1339	10.65	SSR 125543	3
23	+	232.0268	6.97	3-Sulfocatechol	3
24	+	178.0532	1.19	S-Methyl-1-thio-D-glycerate	3
25	−	591.4635	16.27	Bacteriohopanetetrol	3
26	−	282.0194	11.66	Tazobactam	3
27	−	355.0707	7.04	Sulfadoxine	3
28	−	228.9942	9.57	DCEBIO	3
29	−	307.0840	10.07	Allamandin	3
30	−	315.0725	7.16	315.0725_7.16	unknown
31	−	361.1257	6.89	Kasugamycin	3
32	−	225.0629	2.69	5-Azacytidine	3
33	−	244.0359	7.14	Selfotel	3

**Table 10 ijms-25-10965-t010:** Metabolomic feature selection datasets employed for the classification tasks.

Dataset	Local Metabolomic Datasets for GBM		
The Number of Cases	214		
The Number of Patients	107		
The Related Tasks	Pre–post-CRT	12-Month OS	20-Month OS
The Number of Positive Cases	107 (post)	170 (alive)	100 (alive)
The Number of Negative Cases	107 (pre)	44 (dead)	114 (dead)
The Number of Total Features	6015		
Cross-Validation Type	5-Fold Stratified CV		
Feature Selection Methods	mRMR, LASSO for Classification		
Feature Weighting Rule	Rank-Based		
Classifier Models	SVM, LR, KNN, RF, AdaBoost, Voting		
Performance Metric	ACC		

## Data Availability

De-identified data, including clinical data associated with the metabolomic data set, will be shared once analyses for outcomes are complete.

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
