# Peer review of "MetaWise: Combined Feature Selection and Weighting Method to Link the Serum Metabolome to Treatment Response and Survival in Glioblastoma"

_ijms, 2024, doi:10.3390/ijms252010965_

Round 1

Reviewer 1 Report

Comments and Suggestions for Authors

The authors proposed a hybrid approach for feature weighting and selection approaches based on rank to study the prognosis of glioblastoma using metabolomic data. Although the studied problem is interesting, I have the following concerns that need to be addressed. Some of major comments:

1. Why does the proposed method work? In other words, the principle or theoretical basis of the proposed method should be analyzed.

2. The manuscript primarily focuses on the statements of results; however, it is equally crucial to provide a theoretical interpretation of these results to enhance understanding.

3. The authors adopted the default parameter settings for each classifier, but in most cases, the model parameters need to be appropriately tuned and optimized for optimal performance, then you may be able to get some interesting results or conclusions.

4. In Tables, there is no explanation of the meanings represented by the highlights of different colors.

 Specific details:

- "pre and post-CRT" -> "pre- and post-CRT"

- The first paragraph on page 4 is missing a right bracket, specifically, "(i.e., …".

- In line165, "where" should be set to align without indentation.

- Full terms of the abbreviation should be written out when it is first used, e.g., FS (in line 153), m/z (in line226), rt (in line 227), etc.

- Variables should be in italic format.

- There is an issue of extra spaces in the article.

- Table 4 is not pretty.

- With regard to the citation of figures in the text, the format is not consistent, e.g., "Fig 1", "Fig. 1(a)", "Figure 1". Please ensure a consistent format.

- "4.2.1. Feature Selection Methods" -> "4.2.2. Feature Selection Methods"

- "4.2.1. Feature Weighting Methods" -> "4.2.3. Feature Weighting Methods"

Author Response

The authors are very grateful to all reviewers for providing comments on the revised manuscript. Thank you kindly for your work on our manuscript and the thoughtful reviews.

The comments given by the editor and reviewers significantly help improve the paper's overall design and presentation. In revising the manuscript, the authors made efforts to incorporate all the comments offered by the reviewers. 

Response to Editor’s Comments 

The authors also thank the editor for pointing out some issues.

We removed four self-citations including the following references in the manuscript as follows:

  1. Tasci, E. and A. Ugur, A novel pattern recognition framework based on ensemble of handcrafted features on images. Multimedia Tools and Applications, 2022: p. 1-24.

  1. Gokalp, O., E. Tasci, and A. Ugur, A novel wrapper feature selection algorithm based on iterated greedy metaheuristic for sentiment classification. Expert Systems with Applications, 2020. 146: p. 113176.

  1. Krauze, A., et al., AI-Driven Image Analysis in Central Nervous System Tumors-Traditional Machine Learning, Deep Learning and Hybrid Models. Journal of biotechnology and biomedicine, 2022. 5(1): p. 1.

  1. Tasci, E., et al., GradWise: a novel application of a rank-based weighted hybrid filter and embedded feature selection method for glioma grading with clinical and molecular characteristics. Cancers, 2023. 15(18): p. 4628.

We also rephrased the sentences from our similar publications from MDPI to decrease the similarity ratio.

Response to Reviewers’ Comments 

=================================================================== =========

Authors’ Response to Reviewer 1

=================================================================== =========

Comment #1:  Why does the proposed method work? In other words, the principle or theoretical basis of the proposed method should be analyzed.

Response #1:  The authors also thank the reviewer for pointing out this constructive advice. We added the rationale, related explanations, and the principle or theoretical basis of the selection of LASSO and mRMR feature selection methods to Section 4.2.1 Proposed Scheme as below :

“ The main principles of the LASSO and mRMR feature selection methods are briefly described as follows:

LASSO is a type of analysis method that performs feature selection and regularization to enhance the prediction performance and interpretability of the statistical models. It applies the L1 norm penalty to the coefficients by shrinking some of the coefficients to exactly zero, effectively performing feature selection by excluding irrelevant features (i.e., sparsity). The selected features are those that contribute most to the learning model, reducing dimensionality and mitigating overfitting. The mRMR (Minimum Redundancy Maximum Relevance) method is a filter-based feature selection technique that aims to identify features most pertinent to the target variable while reducing redundancy (e.g., correlated) among the selected features. mRMR method ensures that the selected features not only predict the targets well but also complement each other by avoiding redundant information.

The hybridization of LASSO (Least Absolute Shrinkage and Selection Operator) and mRMR (Minimum Redundancy Maximum Relevance) feature selection methods works effectively as it leverages the complementary strengths (i.e., regularization and sparsity for LASSO, and balancing feature redundancy and relevance for mRMR) of both techniques to improve the prediction performance and provide the more robust feature selection process and ensure relevant feature diversity by addressing the limitations of each feature selection method separately.”

Comment #2:  The manuscript primarily focuses on the statements of results; however, it is equally crucial to provide a theoretical interpretation of these results to enhance understanding.

Response #2:  The authors also thank the reviewer for pointing out this constructive advice. We added related theoretical interpretation of these results to Section 2.3.1 and Section 2.3.2 as follows:

For Section 2.3.1:

“These results underscore the significance of feature selection in enhancing model performance, especially for models such as LR and SVM. Both LASSO and mRMR offer effective methods for selecting features, but the optimal choice may vary based on the dataset’s characteristics and the specific model employed. Grasping how different feature selection techniques influence model performance is essential for creating robust predictive models.”

For Section 2.3.2:

“To interpret these results theoretically, the situations can be summarized as follows: The soft voting ensemble model and LR consistently show high accuracy across various feature counts, indicating its effectiveness with feature selection. SVM also maintains solid performance, particularly at higher feature counts. RF demonstrates stable performance, highlighting its robustness in handling feature variations. KNN generally shows the lowest initial performance and relatively minor improvements across feature counts. KNN does reach high accuracy for some cases, such as 87.364% accuracy with 8 features for the pre-post CRT status-based metabolomic dataset, but overall, it seems less affected by the number of features compared to other models. AdaBoost has a competitive performance, but it generally doesn't surpass LR. The soft voting ensemble method generally maintains solid accuracy, showcasing its strength in combining different models for improved predictions. As the number of features decreases, many models start to show diminishing returns in performance. For example, LR and SVM continue to achieve good accuracy even with fewer features, whereas KNN tends to struggle more when the feature count is low (i.e., more vulnerable to dimensionality issues).”

Comment #3:  The authors adopted the default parameter settings for each classifier, but in most cases, the model parameters need to be appropriately tuned and optimized for optimal performance, then you may be able to get some interesting results or conclusions.

Response #3:  The authors thank the reviewer for pointing out this constructive advice.

  • We have added explanations to the Section 2.1. Experimental Process as follows:

To provide standardization for our previous similar studies[1-3] for different data types such as molecular, proteomic, and metabolomic data, and as there are many combinations to obtain from different feature subsets, datasets, classification models, and total or rank-ordered weights, and cross-validation, we adopted the same and default parameter settings.

[1]. Tasci, E., et al., RadWise: A Rank-Based Hybrid Feature Weighting and Selection Method for Proteomic Categorization of Chemoirradiation in Patients with Glioblastoma. Cancers, 2023. 15(10): p. 2672.

[2]. Tasci, E., et al. GradWise: A novel application of a rank-based weighted hybrid filter and embedded feature selection method for glioma grading with clinical and molecular characteristics. Cancers 15.18 (2023): 4628.

[3]. Tasci, E., et al., MGMT ProFWise: Unlocking a New Application for Combined Feature Selection and the Rank-Based Weighting Method to Link MGMT Methylation Status to Serum Protein Expression in Patients with Glioblastoma. International Journal of Molecular Sciences, 2024. 25(7): p. 4082.

  • We have also added explanations to the future work section as follows:

“The classification models’ parameters can be appropriately tuned and optimized to obtain optimal performance and new, and interesting results by taking into account computational load and time requirements.”

Comment #4:  In the tables, there is no explanation of the meanings represented by the highlights of different colors.

“Color changes in the tables from red to green represent performance results from the lowest (red) to the highest values (green).”

 Specific details:

- "pre and post-CRT" -> "pre- and post-CRT"

- The first paragraph on page 4 is missing a right bracket, specifically, "(i.e., …".

- In line 165, "where" should be set to align without indentation.

- Full terms of the abbreviation should be written out when it is first used, e.g., FS (in line 153), m/z (in line 226), rt (in line 227), etc.

- Variables should be in italic format.

- There is an issue of extra spaces in the article.

- Table 4 is not pretty.

- With regard to the citation of figures in the text, the format is not consistent, e.g., "Fig 1", "Fig. 1(a)", "Figure 1". Please ensure a consistent format.

- "4.2.1. Feature Selection Methods" -> "4.2.2. Feature Selection Methods"

- "4.2.1. Feature Weighting Methods" -> "4.2.3. Feature Weighting Methods"

Response #4:  The authors also thank the reviewer for pointing out this constructive advice. We corrected them as follows:

In the tables, there is no explanation of the meanings represented by the highlights of different colors. -> We added the explanation for the meaning of different colors as below:

“Color changes in the tables from red to green display performance results from the lowest (red) to the highest values (green).”

  • "pre and post-CRT" -> "pre- and post-CRT" : Corrected
  • The first paragraph on page 4 is missing a right bracket, specifically, "(i.e., …" : Added
  • In line 165, "where" should be set to align without indentation.

Thank you for this correction. In our manuscript, this looks correct.  We can say that this error is due to the production stage error caused by the journal. We will handle this error in the production stage.

  • Full terms of the abbreviation should be written out when it is first used, e.g., FS (in line 153), m/z (in line 226), rt (in line 227), etc. : Corrected
  • Variables should be in italic format : Corrected
  • There is an issue of extra spaces in the article : Corrected
  • Table 4 is not pretty.

Thank you for this suggestion. In our manuscript, this looks correct.  We can say that this error is due to the production stage error caused by the journal. We will handle this error in the production stage.

  • With regard to the citation of figures in the text, the format is not consistent, e.g., "Fig 1", "Fig. 1(a)", "Figure 1". Please ensure a consistent format.

Thank you for this suggestion. We provided a consistent format for the figures (e.g., Figure 1) and corrected them.

  • "4.2.1. Feature Selection Methods" -> "4.2.2. Feature Selection Methods": Corrected
  • "4.2.1. Feature Weighting Methods" -> "4.2.3. Feature Weighting Methods": Corrected

Reviewer 2 Report

Comments and Suggestions for Authors

Dear authors, congratulations on this manuscript which is not complex is fine and interesting, below however some improvements before it can be considered for publication.

1) Authors' names should be spelled out in full.

2) the abstract should state more of the results

3) The introduction should mention at least the new 2021 classification, for this purpose I suggest you read and excite a work that can be useful for the rest of your manuscript as well:  https://doi.org/10.3390/biomedicines12010008

4) diagnosis crosses imaging methods for example MRI is certainly the standard, there are now integrated methods that parallel what you describe do the same on imaging, for example clustering also here I suggest you a work to read and incorporate: https://doi.org/10.3390/brainsci14030296

5) The discussion could be enhanced

6) Conclusions should not incorporate future directions that others should stand at the end of the discussion or in a separate paragraph obviously before the discussions

This is fine for now, I look forward to reading the revised manuscript.

Comments on the Quality of English Language

minor editing needed 

Author Response

=================================================================== =========

Authors’ Response to Reviewer 2

=================================================================== =========

Dear authors, congratulations on this manuscript which is not complex is fine and interesting, below however some improvements before it can be considered for publication.

Comment #1: Authors' names should be spelled out in full.

Response #1: The authors appreciate the positive feedback from the reviewer. The authors thank the reviewer for highlighting this issue. We agree and have now added the authors’ full names to the manuscript as follows:

Tasci Erdal1, Popa Michael1, Zhuge Ying1, Chappidi Shreya1, Zhang Longze1, Cooley Zgela Theresa1, Sproull Mary1, Mackey Megan1, Kates Heather2, Garrett Timothy2 , Camphausen Kevin1, Krauze Andra Valentina1*.

1 Radiation Oncology Branch, Center for Cancer Research,

National Cancer Institute, NIH,

9000 Rockville Pike, Building 10, CRC,

Bethesda, MD 20892, USA

2 Department of Pathology, Immunology and Laboratory Medicine

University of Florida

1395 Center Dr, M641

Gainesville, FL 32610

Comment #2:  The abstract should state more of the results.

Response #2:  The authors thank the reviewer for pointing out this constructive advice. We agree. This is a very significant point.

We have now added the following statement of the results to the abstract as follows:

“The computational results show that the utilized method achieves 96.711%, 92.093%, and 86.910% accuracy rates with 48, 46, and 33 selected features for CRT, 12-month, and 20-month OS-based metabolomic datasets, respectively.”

Comment #3: The introduction should mention at least the new 2021 classification, for this purpose I suggest you read and excite a work that can be useful for the rest of your manuscript as well:  https://doi.org/10.3390/biomedicines12010008

Response #3:  The authors thank the reviewer for pointing out this constructive advice. We have read, added this reference, and cited this publication. We also mentioned about this publication in the introduction section as follows:

“Different studies emphasize the importance of molecular diagnosis in personalizing glioma treatment, and combining molecular insights with these new techniques can enhance the effectiveness of glioma treatments for patients[7, 8]. The most important genes influencing GBM tumor growth consist of EGFR, PTEN, TP53, IDH1, and MGMT mutations[7].”

Added references:

  1. De Simone, M., et al., Advancements in glioma care: focus on emerging neurosurgical techniques. Biomedicines, 2023. 12(1): p. 8.
  2. Tasci, E., et al., Hierarchical Voting-Based Feature Selection and Ensemble Learning Model Scheme for Glioma Grading with Clinical and Molecular Characteristics. International Journal of Molecular Sciences, 2022. 23(22): p. 14155.

Comment #4: Diagnosis crosses imaging methods for example MRI is certainly the standard, there are now integrated methods that parallel what you describe do the same on imaging, for example clustering also here I suggest you a work to read and incorporate: https://doi.org/10.3390/brainsci14030296

Response #4:  

The authors thank the reviewer for pointing out this suggestion. In our papers, such as RadWise[1], GradWise[2], or MGMT ProFWise[3] paper, we considered different types of molecular or omics data, such as proteomic data.

[1]. Tasci, E., et al., RadWise: A Rank-Based Hybrid Feature Weighting and Selection Method for Proteomic Categorization of Chemoirradiation in Patients with Glioblastoma. Cancers, 2023. 15(10): p. 2672.

[2]. Tasci, E., et al. GradWise: A novel application of a rank-based weighted hybrid filter and embedded feature selection method for glioma grading with clinical and molecular characteristics. Cancers 15.18 (2023): 4628.

[3]. Tasci, E., et al., MGMT ProFWise: Unlocking a New Application for Combined Feature Selection and the Rank-Based Weighting Method to Link MGMT Methylation Status to Serum Protein Expression in Patients with Glioblastoma. International Journal of Molecular Sciences, 2024. 25(7): p. 4082.

We have added the following explanation to the introduction section as below:

To the introduction section:

“In this study, we focus on only metabolomic data-based biomarker selection from large-scale metabolomic panels to apply our methodology to the specific data type and operations.”

We have read, added this reference, and cited this publication. We also mentioned about this publication in the future work section as follows:

To the future work section:

“Ongoing research will be needed to annotate, link, and validate signals to determine how the metabolism interfaces with tumor resistance and outcomes in GBM. This study illustrates, however, that serum metabolism can provide a promising, noninvasive avenue for future study of this devastating disease. As another future direction of this work, we can also handle imaging data such as MRI using AI-driven methodologies (e.g., classification or clustering[34]) for biomarker discovery for GBM-related tasks.”

Added new reference:

[34]. De Simone, M., et al., Clustering Functional Magnetic Resonance Imaging Time Series in Glioblastoma Characterization: A Review of the Evolution, Applications, and Potentials. Brain Sciences, 2024. 14(3): p. 296.

Comment #5: The discussion could be enhanced.

Response #5:  

This is reasonable. We have now enhanced discussion of the identified significant level 1 compounds and related these to relevant serum and plasma data that involves glioma or specifically GBM to interpret how the compounds identified in this analysis link to existing literature. 4 references were added:

  1. Huang, J., et al., A prospective study of serum metabolites and glioma risk. Oncotarget, 2017. 8(41): p. 70366-70377.
  2. Abrigo, J., et al., Cholic and deoxycholic acids induce mitochondrial dysfunction, impaired biogenesis and autophagic flux in skeletal muscle cells. Biological Research, 2023. 56(1): p. 30.
  3. Boulos, J.C., M.R. Yousof Idres, and T. Efferth, Investigation of cancer drug resistance mechanisms by phosphoproteomics. Pharmacological Research, 2020. 160: p. 105091.
  4. Aboud, O., et al., Profile Characterization of Biogenic Amines in Glioblastoma Patients Undergoing Standard-of-Care Treatment. Biomedicines, 2023. 11(8).

Comment #6: Conclusions should not incorporate future directions that others should stand at the end of the discussion or in a separate paragraph obviously before the discussions.

Response #6:  The authors thank the reviewer for pointing out this constructive advice.

We agree with the reviewer.

  • We have revised and changed the conclusion section as follows:

“This study emphasizes the essential role of feature selection in the analysis of metabolomic data, showing that effective selection techniques can greatly improve model performance and clarity. By pinpointing the most relevant compounds linked to specific biological conditions, we can enhance our understanding of metabolic pathways and disease mechanisms. The proposed method uniquely addresses feature selection and weighting using large-scale metabolomic data measured in the serum of patients with GBM at two-time points per and post SOC CRT. Several promising compounds were identified with the ability to distinguish signal pre-CRT vs. post-CRT with 96.711% accuracy and 12-month overall survival with 92.093% and 20-month overall survival with 86.910%. High-level biological annotation of growing metabolomic data, including mapping to signaling pathways, is still evolving. This data is less mature compared to other omics data types.

Our findings reveal that hybrid feature selection methods surpassed traditional techniques, resulting in improved classification accuracy and more reliable predictive models. This highlights the necessity of using advanced methods that are suited to the complexities of metabolomic data. Looking ahead, we advocate for the integration of feature selection processes into standard metabolomic workflows to support biomarker discovery and promote advancements in personalized medicine.”

  • We have made the future work section, and added the following sentences to this section as follows:

“Ongoing research will be needed to annotate, link, and validate signals to determine how the metabolism interfaces with tumor resistance and outcomes in GBM. This study illustrates, however, that serum metabolism can provide a promising, noninvasive avenue for future study of this devastating disease. The classification models’ parameters can be appropriately tuned and optimized to obtain optimal performance and new and interesting results by taking into account computational load and time requirements. As another future direction of this work, we can also handle imaging data such as MRI using AI-driven methodologies (e.g., classification or clustering[34]) for biomarker discovery for GBM-related tasks.”

Additional Explanations

A Grammarly check has been carried out, and corrections applied.

Round 2

Reviewer 1 Report

Comments and Suggestions for Authors

I appreciate the authors' efforts in addressing the issues raised in the review, specially the modifications made to the manuscript based on the reviewers' feedback. I am satisfied with answers provided to my questions. The revised version is acceptable. Good luck.

Reviewer 2 Report

Comments and Suggestions for Authors

Dear authors, congratulations on the revisions the manuscript is now improved.